# Building Generalist Robot Policy from Pre-trained Visual Representations

## Abstract

In this paper, we investigate the use of vision pre-trained models (PTMs) for developing generalist robot manipulation policies. We study whether embodied policies trained with representations from vision and language PTMs are capable of multi-tasking and overcoming domain gaps. Evaluating a set of off-the-shelf vision PTMs, our first finding is that the commonly used global features are generally inadequate for building multi-task robot manipulation policies, while keeping local features significantly improves in-domain performance and out-of-domain generalizibility. Experiment results show that DINOv2, a model trained on conventional vision datasets, outperforms models explicitly designed for robot learning. To bridge the domain gaps, we further experiment on the effect of augmentation methods on embodied robot policies and few-shot adaptation. On the later case, we propose a novel objective by introducing self-distillation to the objectives of few-shot adaptation. Experiment results show that our approach is compatible with multiple PTMs, improving performance on novel domains when the number of demonstration available is limited.

## 1 Introduction

The design of robot manipulation policies has been transformed by advancements in large pre-trained models (PTMs) in natural language processing and computer vision. The success of foundation models has inspired the development of generalist embodied agents. These agents are designed to understand instructions in natural language, perceive their environment through vision inputs, and take actions to interact with the physical world. In robot manipulation tasks, a generalist robot policy aims to perform a wide range of tasks using a unified model or framework. Additionally, a generalist policy also enables more flexible and efficient deployment to various environments and tasks.

As the most successful foundation models, the use of large language models (LLMs) in building generalist robot manipulation policies has been extensively studied (Zitkovich et al., 2023; Szot et al., 2024). The reasoning and planning capabilities of LLMs enable them to serve as high-level policies that plan macro actions in the language domain (Marza et al., 2024). For embodied agents, natural language provides a concise and environment-invariant description of tasks and surroundings. Therefore, policies built from LLMs often demonstrate decent performance when executing multiple tasks and adapting to unseen tasks in a single environment. For embodied agents that utilize both vision and language inputs, vision plays a critical role in defining their perception of objects and environments. However, the question of whether an embodied policy can generalize to environments with unseen visual attributes remains a significant challenge. Thus, the effectiveness of pre-trained visual representations in generalist agents requires extensive study. In this paper, we build robot manipulation policies using frozen representations from vision PTMs and explore the following questions: (1) If not relying on the predictive power of LLMs, are vision PTMs effective for building a multi-task robot manipulation policy? (2) With "high-quality" features from vision PTMs, can policies trained with pre-trained visual representations effectively generalize to unseen environments? (3) How can we effectively and efficiently bridge the domain gaps between training environments and unseen environments?

Most existing works on robot learning with vision PTMs focus on improving the quality of pre-trained visual representations. These PTMs are benchmarked on whether their representations are effective and efficient for learning a high-performance policy for a **single task**. In this paper, we investigate

whether the representations from vision PTMs can be effectively scaled up to train a **multi-task** robot policy. On the Metaworld (Yu et al., 2019) robot manipulation tasks, we found that commonly-used aggregated visual representations (or global features) are often ineffective for training a multi-task policy, as essential information, such as spatial structure, is lost during feature compression. We demonstrate that using the full representations (or local features) from these models significantly boosts the performance of the multi-task policy. Based on this discovery, we evaluate a set of off-the-shelf vision PTMs within our problem setting. Surprisingly, we found that a PTM trained on conventional image datasets (DINOv2 (Oquab et al., 2024)) outperforms the state-of-the-art PTM (VC-1 (Majumdar et al., 2023)), which was pre-trained on explicitly selected datasets related to robot learning.

In building embodied agents, vision PTMs are also termed as artificial visual cortex (Majumdar et al., 2023). As humans, we possess a structured and, in some sense, symbolic understanding of perceived objects and environments, allowing us to easily generalize skills learned from tasks in one domain to similar tasks in novel domains. For example, after learning to drive a white sedan in driving school, we can naturally drive a red sedan on a highway or a road in a forest. We could probably also learn to drive a black SUV with just a few minutes of practice. Similarly, upon seeing a red object, we can immediately grasp the concept of the same object in green. This knowledge of visual representations in the human visual cortex enables us to generalize skills effectively without requiring extensive training on diverse experiences for a specific task. Huh et al. (2024) hypothesize that vision and language PTMs trained on large-scale data converge to representations that are similarly distributed, suggesting they gravitate towards a statistical model of the world. Given that PTMs are trained on internet-scale datasets, could their inductive biases induce a similar pattern when training embodied agents with their representations? In this work, we investigate whether robot manipulation policies trained with PTM representations and demonstrations from a single domain can generically generalize to multiple unseen domains.

When deploying embodied agents, the gap between training task domains and unseen task domains still presents challenges, particularly if the diversity of training domains is limited. Observing that policies trained with vision PTMs usually result in some level of performance degrade on unseen domains, we further investigate possible approaches to bridge these domain gaps. Existing works address domain gaps in robot learning through two main directions: data augmentation (Laskin et al., 2020; Yu et al., 2023) and generation (Tobin et al., 2017; Yang et al., 2024), or few-shot adaptation (Marza et al., 2024). In this work, we explore both directions within our problem setup. First, we assess whether conventional augmentation methods effectively reduce the generalization gap, and we observe that each vision PTM is compatible with different types of augmentation. Next, in the few-shot adaptation setting, we propose a novel approach by introducing self-distillation into the fine-tuning objective.

Our contributions are summarized as follows: (1) We found that the commonly-used global features from vision PTMs are generally ineffective for building multi-task robot manipulation policies, while policies trained with local features from the PTMs achieve significantly better performance. (2) We evaluate a set of existing vision PTMs, comparing their in-domain multi-task performance and out-of-domain generalization, and conclude that policies trained with local features from DINOv2 perform the best on both metrics. (3) We conduct an extensive study on the effects of conventional data augmentation methods on robot policy training with pre-trained visual representations, summarizing the compatibility of augmentation methods with different vision PTMs. (4) We propose a novel objective for few-shot adaptation by introducing self-distillation on features from a trained policy, which improves performance when the number of novel demonstrations is limited and generally outperforms conventional fine-tuning methods when evaluated on unseen domains.

## 2 RELATED WORKS

**Generalist Robot Policy** Two primary research directions have emerged for building generalist robot policies. The first direction leverages the power of large language models (LLMs). Myers et al. (2024) decompose complex tasks into subtasks and use GPT4o to plan the sequence of subtasks for execution. Szot et al. (2024) and Zitkovich et al. (2023) adapt LLMs into vision-language robot policies by mapping visual representations and actions to the embeddings of a frozen LLM. The second direction focuses on creating more generalist robot policies, where vision and language pre-trained models

(PTMs) serve only as feature extractors for visual and language inputs (Brohan et al., 2023). Octo Model Team et al. (2024) trains a Transformer (Vaswani et al., 2017) policy on 800k episodes of robot manipulation tasks, establishing a foundation model for robot manipulation policies. In this paper, we explore the second direction and emphasize the role of vision PTMs in training generalist robot manipulation policies.

**Pre-trained Visual Representations for Downstream Policy Training** Utilizing pre-trained models for downstream tasks is already common in computer vision and natural language processing. However, due to the large domain gap between standard vision benchmarks and control tasks, such strategies have only recently been explored. Parisi et al. (2022) use outputs from multiple layers of a frozen pre-trained MoCo ResNet (He et al., 2020) to train a single-task policy. Shridhar et al. (2021) propose a two-stream framework that employs pre-trained CLIP (Radford et al., 2021) to guide the training of a Transporter model for affordance prediction tasks. Khandelwal et al. (2022) utilize local tokens from the pre-trained CLIP model to perform navigation tasks. Later studies found that favoring egocentric view data in the pre-training distribution improves downstream single-task policy performance. Specifically, R3M uses temporal-contrastive learning with video-text pairs from the Ego4D dataset (Grauman et al., 2022) to enhance single-task policy training. VC-1 (Majumdar et al., 2023) adopts a masked image modeling objective with a diverse dataset to provide unified visual representations for downstream policy training.

**Domain Generalization and Adaptation** A common approach to improve domain generalization is to increase the amount of training data. Yu et al. (2023) use a text-to-image diffusion-based in-painting model to randomly augment objects of interest, selected by an open-vocabulary segmentation model, to enhance domain generalization. Wang et al. (2024) and Yang et al. (2024) further explore generative modeling as a simulator to generate infinite examples. While the ability to produce numerous objects with varying attributes is beneficial, challenges arise due to computational overhead and inconsistencies in object generation across frames.

Marza et al. (2024) propose training a multitask embedding space that controls the output of a pre-trained vision backbone using lightweight adapters. These adapters, along with the embedding space, enable rapid adaptation to new tasks with only a few demonstrations. Recent work (Myers et al., 2024) harnesses the generalization abilities of large vision-language models (such as GPT4o) to generate hierarchical language instructions for adapting to new long-horizon tasks. However, the language model tends to generalize low-level instructions (e.g., referring to both a potato and a turnip toy as "purple thing"), and the objects remain unchanged between training and testing phases.

## 3 PRELIMINARIES

A robot manipulation task $T = (Z, V, G)$ is defined by the natural language instruction $Z$, the image(s) of initial condition $V$, and the goal condition $G$. Then, the language PTM encodes the instruction with $z = \text{PTM}^z(Z)$ where $z \in \mathbb{R}^{d_{\text{lang}}}$ is the instruction feature. The vision PTM encodes the image input with $(v^{\text{global}}, v^{\text{local}}) = \text{PTM}^v(V)$ where $v^{\text{global}} \in \mathbb{R}^{d^{\text{global}}}$ is the global feature vector and $v^{\text{local}} \in \mathbb{R}^{d^{\text{local}}}$ is the local feature map. The policy $\hat{a}_t = \pi(z, v_{t-h+1:t})$ takes the instruction feature and a short history of observations with length of $h$ to predict the action $a_t \in \mathbb{R}^{d^a}$. It is important to note that our problem setup differs from those studied by Nair et al. (2022), Majumdar et al. (2023), and Marza et al. (2024). In our case, the policy does not rely on proprioceptive signals, as these can exhibit strong correlations with actions and goals (Octo Model Team et al., 2024). Thus, we omit proprioceptive signals to focus solely on the effectiveness of visual representations.

A multi-task domain $\mathbb{T} = \{T_1, \dots, T_K\}$ contains $K$ different types of tasks. To test the generalizability, we train the policies on a datasets from a single source domain $\mathbb{T}^{\text{train}}$ and evaluate them on 10 different target domains $\{\mathbb{T}^{\text{test}}_1, \dots \mathbb{T}^{\text{test}}_{10}\}$. Our problem setting differs from the approaches studied by Tobin et al. (2017), Shridhar et al. (2021), and Lin et al. (2024), where policies are trained on **a diverse set of objects** and evaluated on unseen objects. We assume that each task $T_k \in \mathbb{T}^{\text{train}}$ contains **only a single set of objects** while the target domains contain sets of unseen objects. For example, Shridhar et al. (2021) and Lin et al. (2024) assume that $T_k \in \mathbb{T}^{\text{train}}$ includes an object with multiple colors (e.g., red, green, blue, yellow, brown, gray, cyan), and the same type of task $T_k \in \mathbb{T}^{\text{test}}$ involves the object in unseen colors (e.g., orange, purple, pink, white). In our case, we assume $T_k \in \mathbb{T}^{\text{train}}$

only contains the object with a single color {e.g. blue}. In this paper, we refer to $\mathbb{T}^{\text{train}}$ as in-domain tasks and $\{\mathbb{T}^{\text{test}}_1, \ldots \mathbb{T}^{\text{test}}_{10}\}$ as out-of-domain tasks.

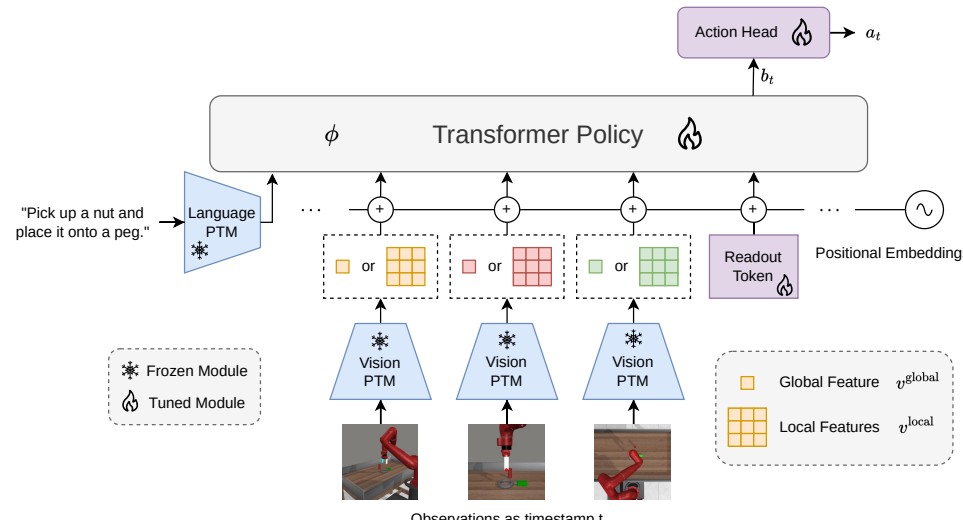

Figure 1: Overview of our multi-tasks policy architecture. Vision PTM provide either global or local features from input images and a language PTM encodes tasks instruction into a instruction token. Action head outputs the final action signal to control the robot.

We use a Transformer policy, shown in Figure 1, that follows the design of Octo (Octo Model Team et al., 2024) but with a deterministic action head. Based on the policy structure in Nair et al. (2022), we incorporate observations from three different cameras—corner view, top view, and gripper view—to reduce the number of partially observable cases where objects of interest are not visible from a single perspective. We select a context window with a horizon of $h = 5$. Following the imitation learning procedure, we train the policy using demonstrations collected with the default expert policy for Metaworld (refer to Appendix A for details). Let $\tau = (Z, V_{1:T}, a_{1:T})$ denotes a demonstration with instruction Z, vision recordings over $T$ timestamps $V_{1:T}$, and action recordings $a_{1:T}$. We denote the training dataset with $N$ demonstrations as $\mathbb{D}^{\text{train}} = \{\tau^n = (Z^n, V^n_{1:T}, a^n_{1:T})|n = 1, \ldots, N\}$.

Key differences between our problem setting and those in prior works (Majumdar et al., 2023; Marza et al., 2024) are: (1) we remove proprioceptive signals to prevent policies from focusing on them instead of visual features, (2) we incorporate three views to minimize the occurrence of partially observable situations, and (3) we consider a one-to-many domain generalization setup, in contrast to the typical many-to-many setting (Tobin et al., 2017; Lin et al., 2024). With these formulations, visual representations play a central role in policy learning, enabling us to compare the effectiveness of features from different PTMs for training robot policies.

## 4 ARE EXISTING VISION PTMS EFFECTIVE FOR TRAINING A GENERALIST ROBOT MANIPULATION POLICY?

In this section, we discuss the effectiveness of pre-trained visual representations in building generalist robot policies. We evaluate a set of off-the-shelf vision PTMs by training policies with their representations and comparing their performance. Table 1 summarizes the key information about the PTMs evaluated in this study. These PTMs utilize various backbones and produce $v^{\text{local}}$ with different spatial dimensions. To ensure a fair comparison, we unify the spatial dimensions of $v^{\text{local}}$ fed into the policy to $7 \times 7$ using adaptive average pooling for models with larger dimensions, such as VC-1 and DINOv2. For CLIP-ViT32 and CLIP-RN50, the text encoder $\text{PTM}^z$ is the paired CLIP text encoder. For all other models, $\text{PTM}^z$ is the frozen DistilBERT (Sanh et al., 2020).

Table 1: Information about the pre-trained vision models studied in this paper.

| Name | Backbone | # Param. | Pre-train Objective | Aggregate Method | $d^{\text{global}}$ | $d^{\text{local}}$ |
|---|---|---|---|---|---|---|
| CLIP-ViT32 | ViT-B/32 | 87.85M | Vision-language Contrastive | CLS Embedding | 512 | $7 \times 7 \times 512$ |
| CLIP-RN50 | ResNet-50 | 38.32M | Vision-language Contrastive | Attention Pooling | 1024 | $7 \times 7 \times 2048$ |
| R3M | ResNet-50 | 23.51M | Vision-language & Temporal Contrastive | Global Average Pooling | 2048 | $7 \times 7 \times 2048$ |
| VC-1 | ViT-B/16 | 85.80M | Masked Image Modelling | CLS Embedding | 768 | $14 \times 14 \times 768$ |
| DINOv2 | ViT-B/14 | 86.58M | Self-distillation | CLS Embedding | 768 | $16 \times 16 \times 768$ |
| DINOv2 (w/ register) | ViT-B/14 | 86.58M | Self-distillation | CLS Embedding | 768 | $16 \times 16 \times 768$ |

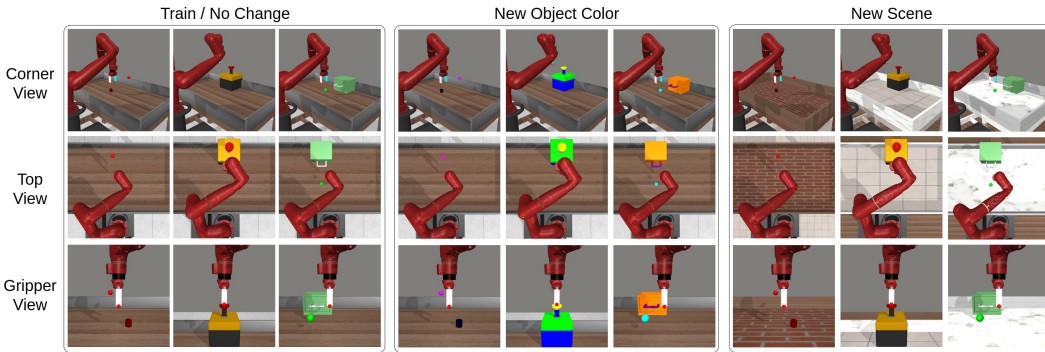

Figure 2: Examples of robot manipulation tasks under different scenarios. **Left**: examples from training and in-domain testing scenario; **Mid**: examples from unseen object colors attributes scenario; **Right**: examples from unseen environment scenario.

### 4.1 PERFORMANCE AND GENERIC GENERALIZIBILITY

A generalist policy should be capable of performing multiple tasks and generalizing to unseen scenarios. Therefore, we benchmark the policies using the following two metrics:

1. **Success rate on in-domain tasks** reflects the policy's ability to imitate expert demonstrations and complete multiple tasks using a single policy. The policies are evaluated on tasks from $\mathbb{T}^{\text{train}}$.

2. **Success rate on out-of-domain tasks** measures the policy's ability to leverage knowledge and skills learned from $\mathbb{T}^{\text{train}}$ to complete tasks in unseen domains. The policies are evaluated on tasks from $\{\mathbb{T}^{\text{test}}_1, \ldots \mathbb{T}^{\text{test}}_{10}\}$.

In this paper, we consider two types of unseen domains: unseen object attributes and unseen environments. Figure 2 presents examples of images from the training domain and these two unseen domains.

For each vision PTM, we train two policies: one using the local features $v^{\text{local}}$ and the other using the global features $v^{\text{global}}$. Details of the policy training procedure can be found in Appendix 5.1. Without any pre-processing of the inputs or modifications to the policy, we directly evaluate the trained policies on tasks across the three domains. Figure 3 summarizes the in-domain and out-of-domain performance of these policies. We observe that, for most PTMs, the global features—commonly used in existing works—fail to produce an effective multi-task policy, even for in-domain tasks.

In contrast, policies trained with local features $v^{\text{local}}$ show significant improvement in in-domain success rates, while also demonstrating varying levels of out-of-domain generalizability. Directly utilizing local features allows the policy to adjust the importance of provided features and retain the spatial structure from PTMs. The different training objectives of PTMs may focus on different aspects of visual information. As observed in many downstream applications utilizing PTM features,

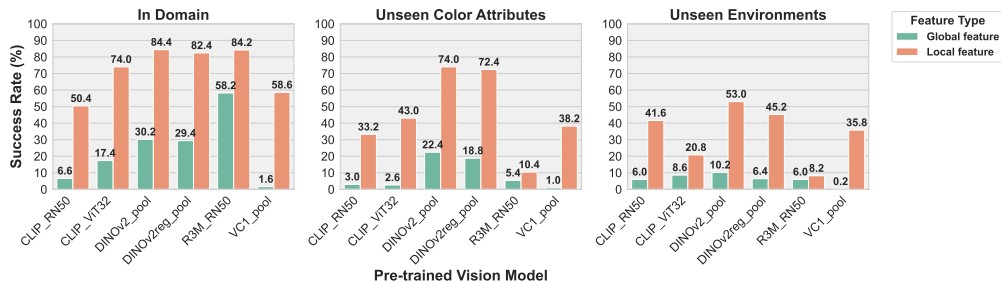

Figure 3: Multi-tasks policy performance (average success rate) comparison between using global feature and local feature using different PTMs under various testing scenarios. **Left**: Evaluated under in-domain environment. **Mid**: Evaluated under environment with unseen object color attributes; **Right**: Evaluated under unseen environments.

a text-image contrastive objective often emphasizes semantic information, while masked image modeling tends to preserve more spatial information. Using local features directly can mitigate the negative impact of inductive biases from PTMs on policy training.

A notable exception is the global feature policy using the R3M backbone, which achieves relatively high performance compared to other PTMs. We speculate several reasons for this: (1) R3M is trained exclusively on the Ego4D dataset, which shares similar viewpoints with the MetaWorld dataset. As observed in Nair et al. (2022), by changing the input view for training single-task policies, only the R3M model maintains a consistent performance ranking compared to other PTMs. (2) The global token of R3M is well-attended in its training objective, whereas models like VC-1 may under-train the global token due to their masked image modeling objective.

Marza et al. (2024) integrate multiple learnable vision adapter layers into the frozen VC-1 backbone to adapt pre-trained features, achieving a $54.5\%$ average success rate on five selected tasks in the MetaWorld dataset. This approach can be seen as a way to reweight pre-trained local features during the forward pass before aggregating them into a global representation. These five tasks are also part of our evaluation dataset. Without any additional modifications, our method, which simply uses local features from the last layer of VC-1, achieves $55.2\%$ on these five subtasks.

From these results, we conclude that: (1) $v^{\text{local}}$ is preferred over $v^{\text{global}}$ when building multi-task robot policies with vision PTMs, (2) with $v^{\text{local}}$, multi-task policies trained with DINOv2 and R3M perform the best on in-domain tasks, and (3) the policy trained with $v^{\text{local}}$ from DINOv2 achieves the best out-of-domain generalizability, suggesting that its essential features may inherently have domain-invariant properties, while the policy trained with $v^{\text{local}}$ from R3M fails to generalize, likely due to overfitting to $\mathbb{T}^{\text{train}}$.

## 4.2 BRIDGING THE DOMAIN GAPS WITH AUGMENTATIONS

The results from previous experiments reveal a significant performance gap between in-domain scenarios and unseen objects/environments, highlighting the limited generalization ability of the policies. To address this, we investigate whether conventional augmentations in pixel space or feature space can enhance the generalization ability of multi-task policies. RAD (Laskin et al., 2020) has demonstrated the effectiveness of augmentations in improving single-policy generalization. Building on this, we propose four different sets of augmentation strategies: pixel-level augmentation, feature noise injection, feature temporal difference, and a mixture of pixel-level and feature noise injection.

**Pixel-level augmentation**: For each example, we randomly select one augmentation from *Random Crop, Random Flip, Random Rotation, Color Jitter, Random Invert, Random Grayscale, and Random Erasing* to augment the input images during training. The performance gains are reported in Table 2. By applying pixel-level augmentation, both CLIP backbones show significant improvements in handling unseen color attributes and environments. However, the R3M backbone experiences a trade-off between performance and generalization when pixel augmentation is applied.

**Feature noise injection augmentation**: We add Gaussian noise to the features from the PTM. The performance is shown in Table 3. The policy trained with the VC-1 backbone benefits the most from this noise injection. While the R3M backbone achieves the highest in-domain performance among all

Table 2: Performance improvement using **pixel** augmentation with local feature. **V**, **C**, **E** stands for in-domain, unseen color attributes and unseen environments respectively. Each cell reports the performance with corresponding augmentation strategy with performance gain compared to policy without augmentation.

| PTMs | V | C | E |
|---|---|---|---|
| CLIP-RN50 | 69.2 (**+18.8**) | 56.6 (**+23.4**) | 65.8 (+24.2) |
| CLIP-ViT32 | 66.8 (-7.2) | 56.4 (+13.4) | 47.2 (**+26.4**) |
| DINOv2-pool | 81.2 (-3.2) | 76.0 (+2.0) | 50.4 (-2.6) |
| DINOv2reg-pool | 72.2 (-10.2) | 69.6 (-2.8) | 50.0 (+4.8) |
| R3M-RN50 | 61.8 (-22.4) | 22.0 (+11.6) | 21.2 (+13.0) |
| VC1-Pool | 54.8 (-3.8) | 50.2 (+12.0) | 41.2 (+5.4) |

Table 3: Performance improvement using **feature noise injection** augmentation with local feature. **V**, **C**, **E** stands for in-domain, unseen color attributes and unseen environments respectively.

| PTMs | V | C | E |
|---|---|---|---|
| CLIP-RN50 | 55.4 (+5.0) | 31.6 (-1.6) | 42.6 (+1.0) |
| CLIP-ViT32 | 72.2 (-1.8) | 42.8 (-0.2) | 26.0 (+5.2) |
| DINOv2-pool | 85.0 (+0.6) | 69.8 (-4.2) | 49.2 (-3.8) |
| DINOv2reg-pool | 79.0 (-3.4) | 73.2 (+0.8) | 51.2 (**+6.0**) |
| R3M-RN50 | 87.2 (+3.0) | 10.2 (-0.2) | 3.6 (-4.6) |
| VC1-pool | 69.4 (**+10.8**) | 44.8 (**+6.6**) | 41.8 (**+6.0**) |

Table 4: Performance improvement using mixture of **pixel** and **feature noise injection** augmentation with local feature. **V**, **C**, **E** stands for in-domain, unseen color attributes and unseen environments respectively.

| PTMs | V | C | E |
|---|---|---|---|
| CLIP-RN50 | 67.0 (**+16.6**) | 57.4 (**+24.2**) | 62.2 (+20.6) |
| CLIP-ViT32 | 65.2 (-8.8) | 53.4 (+10.4) | 46.8 (**+26.0**) |
| DINOv2-pool | 81.8 (-2.6) | 75.2 (+1.2) | 47.6 (-5.4) |
| DINOv2reg-pool | 77.4 (-5.0) | 72.4 (+0.0) | 52.4 (+7.2) |
| R3M-RN50 | 65.6 (-18.6) | 21.4 (+11.0) | 13.2 (+5.0) |
| VC1-Pool | 54.4 (-4.2) | 50.4 (+12.2) | 41.4 (+5.6) |

Table 5: Performance improvement using **temporal difference** augmentation with local feature. **V**, **C**, **E** stands for in-domain, unseen color attributes and unseen environments respectively.

| PTMs | V | C | E |
|---|---|---|---|
| CLIP-RN50 | 58.4 (+8.0) | 32.6 (-0.6) | 49.0 (+7.4) |
| CLIP-ViT32 | 66.2 (-7.8) | 39.4 (-3.6) | 35.8 (**+15.0**) |
| DINOv2-pool | 86.8 (+2.4) | 75.8 (+1.8) | 53.0 (+0.0) |
| DINOv2reg-pool | 83.2 (+0.8) | 71.4 (-1.0) | 45.8 (+0.6) |
| R3M-RN50 | 84.6 (+0.4) | 9.6 (-0.8) | 11.2 (+3.0) |
| VC1-Pool | 81.8 (**+23.2**) | 49.2 (**+11.0**) | 36.0 (+0.2) |

models, its generalization ability is further degraded. We also combine feature-level and pixel-level augmentations, and the results are presented in Table 4.

**Temporal difference augmentation**: This strategy involves subtracting frame features from the first frame's feature within the horizon. Although the policy using the DINOv2 backbone already achieves a high success rate, adding temporal difference augmentation further boosts its performance without any degradation. We also report the performance gains using global features in Tables 6, 7, 8, and 9, showing similar results.

In this section, we experiment with different augmentation strategies without introducing additional data. Incorporating augmentation during training does improve generalization to some extent. Another important consideration is how we can quickly enhance the generalization ability of a well-performing multi-task policy with a small amount of new data (e.g., one demonstration) of unseen objects and environments. In the next section, we propose an efficient method that quickly adapts to unseen objects and environments with limited demonstrations, without sacrificing high in-domain performance.

## 5 FEW-SHOT ADAPTATION

In this section, we introduce a sample-efficient method for adapting a trained policy to unseen domains in a few-shot setting. Given a policy $\pi^{\text{train}}$, trained on demonstrations from $\mathbb{D}^{\text{train}}$, our goal is to adapt this policy to unseen domains by learning from only a few demonstrations $\mathbb{D}^{\text{ft}}$. Existing methods for few-shot adaptation typically imitate actions from the demonstrations using various techniques. A common approach involves fine-tuning the policy or just the action head (Octo Model Team et al., 2024). Marza et al. (2024) propose searching for a task embedding that controls the intermediate features of the vision PTM. In this paper, we present a novel approach by introducing feature distillation into the fine-tuning objective. To the best of our knowledge, we are the first to incorporate self-distillation techniques in domain adaptation for robot manipulation policies.

Self-distillation methods in computer vision typically align the features of two augmented views of an image (Grill et al., 2020; Zhou et al., 2022). Inspired by these approaches, our method aligns the features of two demonstrations that exhibit similar behavior. Given the policy $\pi^{\text{train}}$, the training

Table 6: Performance improvement using **pixel** augmentation with global feature. **V**, **C**, **E** stands for in-domain, unseen color attributes and unseen environments respectively. Each cell reports the performance with corresponding augmentation strategy with performance gain compared to policy without augmentation.

| PTMs | V | C | E |
|---|---|---|---|
| CLIP-RN50 | 30.2 (**+23.6**) | 28.0 (+25.0) | 30.2 (**+24.2**) |
| CLIP-ViT32 | 39.8 (+22.4) | 31.0 (**+28.4**) | 20.0 (+11.4) |
| DINOv2-pool | 39.2 (+9.0) | 37.8 (+15.4) | 30.2 (+20.0) |
| DINOv2reg-pool | 32.8 (+3.4) | 29.6 (+10.8) | 20.0 (+13.6) |
| R3M-RN50 | 32.2 (-26.0) | 9.0 (+3.6) | 1.4 (-4.6) |
| VC1-Pool | 6.8 (+5.2) | 14.8 (+13.8) | 9.2 (+9.0) |

Table 7: Performance improvement using **feature noise injection** augmentation with global feature. **V**, **C**, **E** stands for in-domain, unseen color attributes and unseen environments respectively.

| PTMs | V | C | E |
|---|---|---|---|
| CLIP-RN50 | 7.8 (+1.2) | 5.2 (+2.2) | 4.8 (-1.2) |
| CLIP-ViT32 | 8.2 (-9.2) | 4.2 (+1.6) | 5.8 (-2.8) |
| DINOv2-pool | 31.0 (+0.8) | 26.8 (+4.4) | 14.8 (**+4.6**) |
| DINOv2reg-pool | 21.2 (-8.2) | 14.0 (-4.8) | 9.4 (+3.0) |
| R3M-RN50 | 57.6 (-0.6) | 5.2 (-0.2) | 4.0 (-2.0) |
| VC1-pool | 10.0 (**+8.4**) | 7.0 (**+6.0**) | 2.8 (+2.6) |

Table 8: Performance improvement using mixture of **pixel** and **feature noise injection** augmentation with global feature. **V**, **C**, **E** stands for in-domain, unseen color attributes and unseen environments respectively.

| PTMs | V | C | E |
|---|---|---|---|
| CLIP-ViT32 | 26.8 (+20.2) | 28.6 (+25.6) | 24.8 (+18.8) |
| CLIP-ViT32 | 39.6 (**+22.2**) | 31.6 (**+29.0**) | 20.8 (+12.2) |
| DINOv2-pool | 39.8 (+9.6) | 31.2 (+8.8) | 21.2 (+11.0) |
| DINOv2reg-pool | 32.8 (+3.4) | 30.6 (+11.8) | 21.6 (+15.2) |
| R3M-RN50 | 31.2 (-27.0) | 16.4 (+11.0) | 3.8 (-2.2) |
| VC1-Pool | 23.0 (+21.4) | 20.4 (+19.4) | 19.8 (**+19.6**) |

Table 9: Performance improvement using **temporal difference** augmentation with global feature. **V**, **C**, **E** stands for in-domain, unseen color attributes and unseen environments respectively.

| PTMs | V | C | E |
|---|---|---|---|
| CLIP-RN50 | 4.8 (-1.8) | 4.0 (+1.0) | 6.6 (+0.6) |
| CLIP-ViT32 | 20.4 (+3.0) | 4.0 (+1.4) | 9.2 (+0.6) |
| DINOv2-pool | 40.2 (**+10.0**) | 33.6 (**+11.2**) | 21.8 (**+11.6**) |
| DINOv2reg-pool | 38.0 (+8.6) | 29.0 (+10.2) | 16.2 (+9.8) |
| R3M-RN50 | 57.0 (-1.2) | 2.6 (-2.8) | 3.0 (-3.0) |
| VC1-Pool | 2.8 (+1.2) | 2.4 (+1.4) | 3.8 (+3.6) |

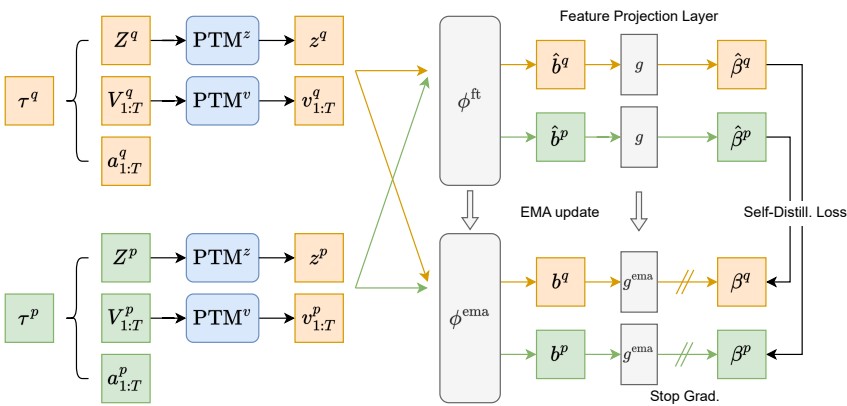

Figure 4: Visualization for procedures of self-distillation.

dataset $\mathbb{D}^{\text{train}}$, and a query demonstration from the unseen domain $\tau^q = (Z^q, V_{1:T}^q, a_{1:T}^q) \in \mathbb{D}^{\text{ft}}$, we first identify the demonstration $\tau^p \in \mathbb{D}^{\text{train}}$ that has the most similar instruction and action recordings to $\tau^q$, and treat them as paired demonstrations.

When the number of demonstrations in $\mathbb{D}^{\text{ft}}$ is limited, fine-tuning $\pi^{\text{train}}$ using behavior cloning may lead to overfitting on $\mathbb{D}^{\text{ft}}$, without proper generalization to $\mathbb{T}^{\text{test}}$. By using paired demonstrations, we can account for the domain shift between $\mathbb{T}^{\text{train}}$ and $\mathbb{T}^{\text{test}}$ during fine-tuning. Building on this idea, we employ a self-distillation approach that adds an extra term to the fine-tuning objective, reducing overfitting by aligning the features across domains. Figure 4 illustrates our proposed self-distillation approach. Here, $\phi$ represents the Transformer component in policy $\pi$, which processes the input features and outputs the action embedding $b$. $g$ is a learnable projection layer, with its outputs $\beta$ normalized using `softmax` (Zhou et al., 2022). In self-distillation terminology, $\phi^{\text{ft}}$ is the student model, and $\phi^{\text{ema}}$ is the teacher model, whose parameters are updated through an Exponential Moving Average (EMA) from the student model's parameters.

Formally, our proposed adaptation method by fine-tuning the policy with the follow objective:

$$
\mathcal{L}_{\text{ft}} = \underbrace{\sum_{\tau^{\text{ft}} \in \mathbb{D}^{\text{ft}}} \sum_{t=h}^{T} |a_t^{\text{ft}} - \pi^{\text{ft}}(z^{\text{ft}}, v_{t-h+1:t}^{\text{ft}})|^2}_{\text{(i) behavior cloning on few-shot domains}} + \underbrace{\sum_{\tau^{\text{train}} \in \mathbb{D}^{\text{train}}} \sum_{t=h}^{T} |a_t^{\text{train}} - \pi^{\text{ft}}(z^{\text{train}}, v_{t-h+1:t}^{\text{train}})|^2}_{\text{(ii) behavior cloning on training domains}}
$$

$$
+ \underbrace{\lambda^{\text{distill}} \sum_{(\tau^q, \tau^p) \in (\mathbb{D}^{\text{ft}}, \mathbb{D}^{\text{train}})} \sum_{t=h}^{T} \texttt{KLDiv}(\hat{\beta}^q, \beta^p) + \texttt{KLDiv}(\hat{\beta}^p, \beta^q)}_{\text{(iii) self-distillation}},
$$

(1)

where $\texttt{KLDiv}(\beta^q, \beta^q)$ is the KL-Divergence of distributions $\beta^q$ and $\beta^q$. Intuitively, component (i) optimizes the out-of-domain performance using the few-shot demonstrations while component (ii) retains the performance on in-domain tasks; component (iii) optimizes the action embeddings to account for domain shift between $\mathbb{D}^{\text{ft}}$ and $\mathbb{D}^{\text{train}}$.

## 5.1 EXPERIMENT RESULTS ON FEW-SHOT ADAPTATION

For benchmarking and fair comparison, we maintain the same configurations across all experiments in this paper. The detailed experimental setups are provided in Table 10 of Appendix A. Detailed experiment results are included in Appendix B.

The training task domain $\mathbb{T}^{\text{train}}$ consists of 10 tasks from the Metaworld benchmark, using datasets selected by Yu et al. (2019) and Majumdar et al. (2023). These tasks are: assembly, bin-picking, button-press-topdown, door-open, drawer-open, hammer, pick-place, push, reach, and window-open. The training dataset $\mathbb{D}^{\text{train}}$ contains 500 expert demonstrations, with 50 demonstrations per task. During evaluation, an episode ends either when the goal condition $G$ is reached (success) or when the maximum step limit is reached (failure).

The evaluation task domains consist of 5 domains with unseen object colors and 5 domains with unseen environments. Each $\mathbb{T}^{\text{test}}$ includes the same 10 tasks as $\mathbb{T}^{\text{train}}$, but with randomized initial conditions. For each $T \in \mathbb{T}^{\text{test}}$, we evaluate the policy 10 times with 10 different random initial conditions and report the average success rates.

In the few-shot adaptation settings, we evaluate four representative PTMs: CLIP-ViT32, R3M, VC-1, and DINOv2. We experiment with varying numbers of demonstrations {1, 2, 5} per task in the test task domains. We benchmark our proposed self-distillation method against two baselines: (1) Baseline: the success rate of $\pi^{\text{train}}$, and (2) Fine-tuning: the success rate of $\pi^{\text{ft}}$, fine-tuned with only components (i) and (ii) from Equation 1.

Figures 5, 6, 7, and 8 compare our approach with the baseline and conventional fine-tuning across the four PTMs. When evaluated on unseen environments, our approach consistently improves the performance of fine-tuned policies, especially when only 1 or 2 demonstrations are available per task. We observe similar, though less pronounced, improvements when evaluated on unseen tasks with novel object colors. The most significant performance gains are observed in policies trained with CLIP-ViT32. In other cases, our approach maintains performance comparable to conventional fine-tuning.

When 5 demonstrations are available for each task, the fine-tuning dataset $\mathbb{D}^{\text{ft}}$ contains 500 samples, which is sufficient to capture the domain gaps comprehensively. In these instances, adding the self-distillation term does not yield further performance improvements. We conclude that our proposed method effectively enhances performance when the number of demonstrations from unseen domains is limited.

## 6 CONCLUSION

In this paper we investigate effective ways of building multi-tasks policies using vision PTMs. By carefully evaluating in-domain and out-of-domain generalization ability of trained policy, we find simply keeping local features from the last layers of PTMs can significantly improve the policy performance compared to the global feature counterpart that is widely used for policy training.

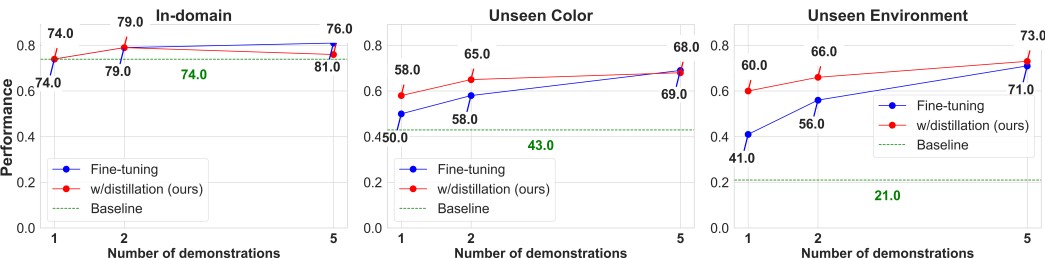

Figure 5: Few-shot adaptation: Success rate of $\pi^{\text{ft}}$ trained with CLIP-ViT32

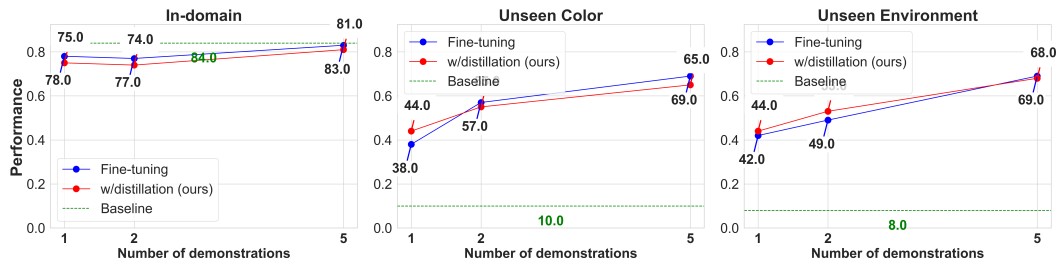

Figure 6: Few-shot adaptation: Success rate of $\pi^{\text{ft}}$ trained with R3M-RN50

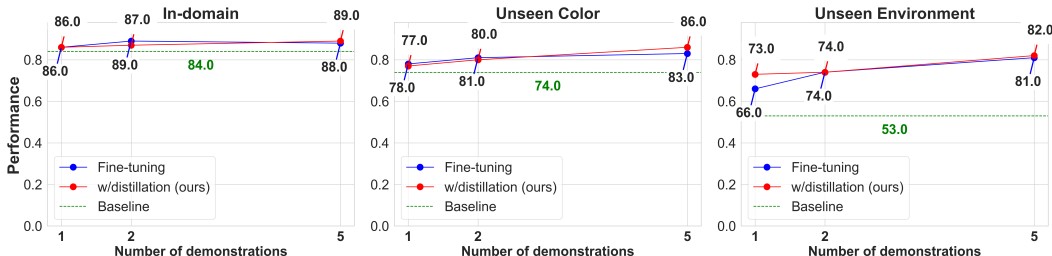

Figure 7: Few-shot adaptation: Success rate of $\pi^{\text{ft}}$ trained with DINOv2-pool

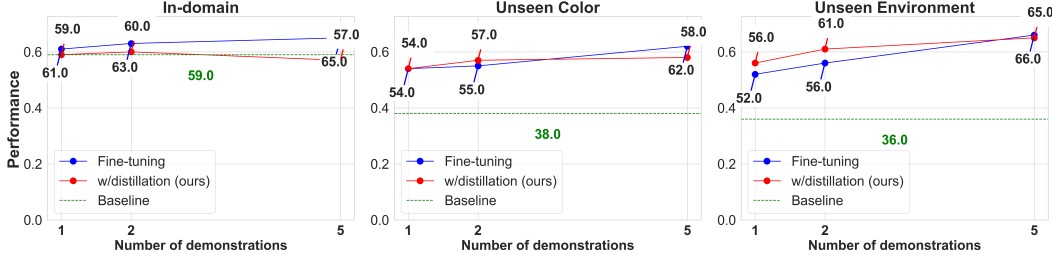

Figure 8: Few-shot adaptation: Success rate of $\pi^{\text{ft}}$ trained with VC1-pool

This finding can simplify the way of utilizing vision PTMs for policy training while achieving high performance. Further, we explored different perspectives of improving policy generalization ability. From the augmentation perspective, we observed policies using different PTM's have clear preference in augmentation strategies. It is challenging to come up with a unified augmentation pipeline for training policies using different PTMs. On the other hand, we propose a novel objective that is able to quickly improve generalization ability under few-shot setting. This method provides overall improvements for policy with different PTMs in unseen scenarios. Our work also has several limitations: (1) our proposed method can not further boost the policy performance when the number of samples increases compared to weighted fine-tuning. (2) The performance is sensitive to teacher model's updating factor. We plan to improve these aspects in future work.

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

## A EXPERIMENT SETUP

Table 10: Experiment Configurations

| Hyperparameter | Value |
|---|---|
| Number of policy layers | 4 |
| Number of attention heads | 8 |
| Policy embedding dimension | 512 |
| Dropout | 0.1 |
| Train epochs | 30 |
| Train learning rate | $3 \cdots 10^{-4}$ |
| Train learning rate schedule | Linear warmup with cosine decay |
| Warmup epoch | 1 |
| Gradient clip norm | 1.0 |
| Weight decay | 0.01 |
| Batch size | 64 |
| Context window length $h$ | 5 |
| Few-shot adaptation epoch | 10 |
| Few-shot adaptation learning rate | $1 \cdots 10^{-4}$ |
| Few-shot adaptation learning rate schedule | Linear warmup with cosine decay |

## B FULL EXPERIMENT RESULTS

Table 11: Success rate of policies trained with local features from CLIP-ViT32

| Augmentation | In-domain | | | | | Unseen Color Attributes | | | | | Unseen Environments | | | | |
|---|---|---|---|---|---|---|---|---|---|---|---|---|---|---|---|
| | none | pixel | feature | p.+f. | t.d. | none | pixel | feature | p.+f. | t.d. | none | pixel | feature | p.+f. | t.d. |
| assembly | 0.60 | 0.76 | 0.40 | 0.60 | 0.36 | 0.14 | 0.58 | 0.14 | 0.46 | 0.28 | 0.06 | 0.22 | 0.00 | 0.44 | 0.00 |
| bin-picking | 0.28 | 0.42 | 0.60 | 0.04 | 0.28 | 0.00 | 0.58 | 0.00 | 0.24 | 0.00 | 0.00 | 0.34 | 0.00 | 0.22 | 0.00 |
| button-press-topdown | 0.76 | 0.86 | 0.80 | 0.82 | 0.90 | 0.80 | 0.90 | 0.60 | 0.88 | 0.68 | 0.52 | 0.72 | 0.80 | 0.66 | 0.86 |
| door-open | 1.00 | 1.00 | 1.00 | 1.00 | 0.98 | 0.90 | 0.92 | 1.00 | 0.98 | 0.96 | 0.24 | 1.00 | 0.24 | 0.94 | 0.46 |
| drawer-open | 1.00 | 0.96 | 1.00 | 1.00 | 1.00 | 1.00 | 1.00 | 1.00 | 0.90 | 0.76 | 0.26 | 0.92 | 0.36 | 0.50 | 0.88 |
| hammer | 0.96 | 0.78 | 0.90 | 0.84 | 0.72 | 0.32 | 0.56 | 0.40 | 0.52 | 0.22 | 0.18 | 0.36 | 0.32 | 0.34 | 0.30 |
| pick-place | 0.84 | 0.60 | 0.58 | 0.60 | 0.82 | 0.26 | 0.22 | 0.22 | 0.34 | 0.24 | 0.08 | 0.16 | 0.06 | 0.20 | 0.10 |
| push | 0.84 | 0.74 | 0.88 | 0.82 | 0.98 | 0.30 | 0.44 | 0.24 | 0.36 | 0.42 | 0.26 | 0.52 | 0.30 | 0.58 | 0.44 |
| reach | 0.76 | 0.18 | 0.64 | 0.46 | 0.38 | 0.40 | 0.22 | 0.44 | 0.24 | 0.18 | 0.36 | 0.24 | 0.36 | 0.34 | 0.34 |
| window-open | 0.36 | 0.38 | 0.42 | 0.34 | 0.20 | 0.18 | 0.22 | 0.24 | 0.42 | 0.20 | 0.12 | 0.24 | 0.16 | 0.46 | 0.20 |
| Average | 0.74 | 0.67 | 0.72 | 0.65 | 0.66 | 0.43 | 0.56 | 0.43 | 0.53 | 0.39 | 0.21 | 0.47 | 0.26 | 0.47 | 0.36 |

Table 12: Success rate of policies trained with local features from CLIP-RN50

| Augmentation | In-domain | | | | | Unseen Color Attributes | | | | | Unseen Environments | | | | |
|---|---|---|---|---|---|---|---|---|---|---|---|---|---|---|---|
| | none | pixel | feature | p.+f. | t.d. | none | pixel | feature | p.+f. | t.d. | none | pixel | feature | p.+f. | t.d. |
| assembly | 0.30 | 0.82 | 0.74 | 0.72 | 0.12 | 0.08 | 0.76 | 0.04 | 0.58 | 0.00 | 0.02 | 0.74 | 0.06 | 0.68 | 0.08 |
| bin-picking | 0.26 | 0.68 | 0.36 | 0.40 | 0.46 | 0.00 | 0.02 | 0.00 | 0.00 | 0.00 | 0.34 | 0.64 | 0.30 | 0.46 | 0.44 |
| button-press-topdown | 0.62 | 0.98 | 0.54 | 1.00 | 0.92 | 0.16 | 0.94 | 0.10 | 0.96 | 0.24 | 0.40 | 0.96 | 0.46 | 1.00 | 0.88 |
| door-open | 0.92 | 1.00 | 0.90 | 0.96 | 1.00 | 1.00 | 1.00 | 0.98 | 1.00 | 1.00 | 0.98 | 0.98 | 1.00 | 1.00 | 1.00 |
| drawer-open | 1.00 | 1.00 | 1.00 | 1.00 | 1.00 | 0.68 | 1.00 | 1.00 | 1.00 | 1.00 | 1.00 | 0.96 | 1.00 | 1.00 | 0.90 |
| hammer | 0.24 | 0.98 | 0.22 | 1.00 | 0.28 | 0.16 | 0.90 | 0.10 | 0.88 | 0.20 | 0.00 | 0.62 | 0.02 | 0.48 | 0.10 |
| pick-place | 0.30 | 0.38 | 0.24 | 0.42 | 0.42 | 0.06 | 0.04 | 0.04 | 0.06 | 0.00 | 0.04 | 0.30 | 0.12 | 0.20 | 0.06 |
| push | 0.38 | 0.28 | 0.42 | 0.52 | 0.58 | 0.44 | 0.30 | 0.28 | 0.46 | 0.18 | 0.38 | 0.44 | 0.22 | 0.52 | 0.46 |
| reach | 0.58 | 0.28 | 0.64 | 0.14 | 0.66 | 0.12 | 0.20 | 0.22 | 0.20 | 0.34 | 0.62 | 0.26 | 0.68 | 0.16 | 0.56 |
| window-open | 0.44 | 0.52 | 0.48 | 0.54 | 0.40 | 0.62 | 0.50 | 0.40 | 0.60 | 0.30 | 0.38 | 0.68 | 0.40 | 0.72 | 0.42 |
| Average | 0.50 | 0.69 | 0.55 | 0.67 | 0.58 | 0.33 | 0.57 | 0.32 | 0.57 | 0.33 | 0.42 | 0.66 | 0.43 | 0.62 | 0.49 |

Table 13: Success rate of policies trained with local features from R3M-RN50

| Augmentation | In-domain | | | | | Unseen Color Attributes | | | | | Unseen Environments | | | | |
|---|---|---|---|---|---|---|---|---|---|---|---|---|---|---|---|
| | none | pixel | feature | p.+f. | t.d. | none | pixel | feature | p.+f. | t.d. | none | pixel | feature | p.+f. | t.d. |
| assembly | 0.60 | 0.72 | 0.54 | 0.62 | 0.98 | 0.00 | 0.12 | 0.00 | 0.04 | 0.00 | 0.00 | 0.00 | 0.00 | 0.00 | 0.00 |
| bin-picking | 0.30 | 0.00 | 0.48 | 0.06 | 0.18 | 0.00 | 0.04 | 0.00 | 0.00 | 0.00 | 0.00 | 0.00 | 0.00 | 0.00 | 0.00 |
| button-press-topdown | 1.00 | 0.90 | 1.00 | 0.96 | 1.00 | 0.12 | 0.26 | 0.00 | 0.60 | 0.02 | 0.00 | 0.66 | 0.00 | 0.20 | 0.00 |
| door-open | 1.00 | 1.00 | 1.00 | 1.00 | 0.92 | 0.00 | 0.00 | 0.00 | 0.00 | 0.00 | 0.02 | 0.10 | 0.06 | 0.04 | 0.00 |
| drawer-open | 1.00 | 0.82 | 1.00 | 0.86 | 1.00 | 0.46 | 0.82 | 0.44 | 0.86 | 0.30 | 0.64 | 0.38 | 0.20 | 0.28 | 0.68 |
| hammer | 1.00 | 0.76 | 1.00 | 0.84 | 1.00 | 0.00 | 0.48 | 0.00 | 0.18 | 0.10 | 0.00 | 0.16 | 0.00 | 0.06 | 0.06 |
| pick-place | 1.00 | 0.80 | 1.00 | 0.72 | 1.00 | 0.00 | 0.06 | 0.00 | 0.10 | 0.00 | 0.00 | 0.04 | 0.00 | 0.00 | 0.00 |
| push | 0.98 | 0.90 | 1.00 | 0.90 | 1.00 | 0.04 | 0.22 | 0.02 | 0.20 | 0.16 | 0.02 | 0.16 | 0.02 | 0.16 | 0.04 |
| reach | 0.58 | 0.08 | 0.78 | 0.20 | 0.78 | 0.42 | 0.16 | 0.56 | 0.16 | 0.38 | 0.14 | 0.14 | 0.08 | 0.14 | 0.22 |
| window-open | 0.96 | 0.20 | 0.92 | 0.40 | 0.60 | 0.00 | 0.04 | 0.00 | 0.00 | 0.00 | 0.00 | 0.48 | 0.00 | 0.44 | 0.12 |
| Average | 0.84 | 0.62 | 0.87 | 0.66 | 0.85 | 0.10 | 0.22 | 0.10 | 0.21 | 0.10 | 0.08 | 0.21 | 0.04 | 0.13 | 0.11 |

Table 14: Success rate of policies trained with local features from VC-1-pool

| Augmentation | In-domain | | | | | Unseen Color Attributes | | | | | Unseen Environments | | | | |
|---|---|---|---|---|---|---|---|---|---|---|---|---|---|---|---|
| | none | pixel | feature | p.+f. | t.d. | none | pixel | feature | p.+f. | t.d. | none | pixel | feature | p.+f. | t.d. |
| assembly | 0.10 | 0.48 | 0.60 | 0.90 | 0.90 | 0.14 | 0.52 | 0.46 | 0.80 | 0.60 | 0.00 | 0.12 | 0.06 | 0.36 | 0.14 |
| bin-picking | 0.30 | 0.06 | 0.26 | 0.00 | 0.62 | 0.00 | 0.04 | 0.00 | 0.00 | 0.00 | 0.08 | 0.00 | 0.10 | 0.00 | 0.10 |
| button-press-topdown | 0.80 | 0.90 | 0.96 | 0.80 | 0.98 | 0.40 | 0.88 | 0.38 | 0.82 | 0.20 | 0.74 | 0.90 | 0.90 | 0.84 | 0.92 |
| door-open | 1.00 | 1.00 | 0.94 | 0.92 | 0.98 | 0.32 | 0.98 | 0.18 | 1.00 | 0.16 | 0.80 | 0.96 | 0.70 | 0.92 | 0.40 |
| drawer-open | 1.00 | 1.00 | 1.00 | 1.00 | 1.00 | 0.74 | 1.00 | 0.82 | 1.00 | 0.98 | 0.46 | 0.62 | 0.54 | 0.80 | 0.62 |
| hammer | 0.56 | 0.28 | 0.50 | 0.44 | 0.66 | 0.36 | 0.30 | 0.26 | 0.50 | 0.28 | 0.36 | 0.32 | 0.50 | 0.28 | 0.38 |
| pick-place | 0.74 | 0.46 | 0.98 | 0.38 | 0.98 | 0.54 | 0.26 | 0.58 | 0.16 | 0.70 | 0.08 | 0.22 | 0.06 | 0.02 | 0.04 |
| push | 0.58 | 0.70 | 0.88 | 0.46 | 1.00 | 0.58 | 0.48 | 0.84 | 0.24 | 0.88 | 0.34 | 0.42 | 0.28 | 0.14 | 0.16 |
| reach | 0.20 | 0.14 | 0.38 | 0.12 | 0.68 | 0.28 | 0.26 | 0.48 | 0.22 | 0.68 | 0.30 | 0.18 | 0.48 | 0.20 | 0.32 |
| window-open | 0.58 | 0.46 | 0.44 | 0.42 | 0.38 | 0.46 | 0.30 | 0.48 | 0.30 | 0.44 | 0.42 | 0.38 | 0.56 | 0.58 | 0.52 |
| Average | 0.59 | 0.55 | 0.69 | 0.54 | 0.82 | 0.38 | 0.50 | 0.45 | 0.50 | 0.49 | 0.36 | 0.41 | 0.42 | 0.41 | 0.36 |

Table 15: Success rate of policies trained with local features from DINOv2-pool

| Augmentation | In-domain | | | | | Unseen Color Attributes | | | | | Unseen Environments | | | | |
|---|---|---|---|---|---|---|---|---|---|---|---|---|---|---|---|
| | none | pixel | feature | p.+f. | t.d. | none | pixel | feature | p.+f. | t.d. | none | pixel | feature | p.+f. | t.d. |
| assembly | 0.84 | 0.94 | 0.80 | 0.98 | 0.96 | 0.90 | 0.92 | 0.82 | 0.94 | 0.76 | 0.18 | 0.52 | 0.10 | 0.60 | 0.10 |
| bin-picking | 0.48 | 0.72 | 0.76 | 0.70 | 0.82 | 0.02 | 0.50 | 0.12 | 0.54 | 0.36 | 0.26 | 0.46 | 0.44 | 0.34 | 0.66 |
| button-press-topdown | 0.96 | 0.88 | 1.00 | 0.98 | 1.00 | 0.98 | 0.98 | 1.00 | 1.00 | 0.94 | 0.98 | 0.86 | 1.00 | 0.96 | 0.98 |
| door-open | 0.96 | 1.00 | 0.98 | 1.00 | 1.00 | 1.00 | 1.00 | 1.00 | 1.00 | 1.00 | 0.78 | 0.92 | 0.76 | 0.94 | 1.00 |
| drawer-open | 1.00 | 1.00 | 1.00 | 1.00 | 1.00 | 1.00 | 1.00 | 1.00 | 1.00 | 1.00 | 1.00 | 0.88 | 0.98 | 0.48 | 1.00 |
| hammer | 0.98 | 1.00 | 0.96 | 1.00 | 0.96 | 0.92 | 0.98 | 1.00 | 0.88 | 1.00 | 0.40 | 0.12 | 0.18 | 0.28 | 0.14 |
| pick-place | 0.88 | 0.88 | 0.72 | 0.86 | 0.92 | 0.64 | 0.66 | 0.38 | 0.58 | 0.82 | 0.24 | 0.14 | 0.10 | 0.12 | 0.18 |
| push | 0.86 | 0.98 | 0.86 | 0.94 | 1.00 | 0.72 | 1.00 | 0.82 | 1.00 | 0.92 | 0.40 | 0.68 | 0.30 | 0.52 | 0.54 |
| reach | 0.62 | 0.44 | 0.60 | 0.42 | 0.84 | 0.52 | 0.38 | 0.44 | 0.32 | 0.58 | 0.44 | 0.34 | 0.30 | 0.30 | 0.46 |
| window-open | 0.86 | 0.28 | 0.82 | 0.30 | 0.18 | 0.70 | 0.18 | 0.40 | 0.26 | 0.20 | 0.62 | 0.12 | 0.76 | 0.22 | 0.24 |
| Average | 0.84 | 0.81 | 0.85 | 0.82 | 0.87 | 0.74 | 0.76 | 0.70 | 0.75 | 0.76 | 0.53 | 0.50 | 0.49 | 0.48 | 0.53 |

Table 16: Success rate of policies trained with local features from DINOv2reg-pool

| Augmentation | In-domain | | | | | Unseen Color Attributes | | | | | Unseen Environments | | | | |
|---|---|---|---|---|---|---|---|---|---|---|---|---|---|---|---|
| | none | pixel | feature | p.+f. | t.d. | none | pixel | feature | p.+f. | t.d. | none | pixel | feature | p.+f. | t.d. |
| assembly | 0.62 | 1.00 | 0.52 | 0.96 | 1.00 | 0.48 | 0.96 | 0.16 | 0.96 | 0.82 | 0.04 | 0.14 | 0.22 | 0.16 | 0.02 |
| bin-picking | 0.30 | 0.00 | 0.08 | 0.44 | 0.16 | 0.14 | 0.16 | 0.22 | 0.06 | 0.10 | 0.08 | 0.32 | 0.30 | 0.30 | 0.00 |
| button-press-topdown | 1.00 | 0.94 | 1.00 | 0.82 | 1.00 | 1.00 | 0.98 | 1.00 | 1.00 | 1.00 | 0.98 | 0.92 | 0.96 | 0.76 | 0.98 |
| door-open | 1.00 | 1.00 | 0.98 | 1.00 | 1.00 | 1.00 | 1.00 | 1.00 | 1.00 | 1.00 | 1.00 | 0.94 | 1.00 | 0.92 | 0.96 |
| drawer-open | 1.00 | 1.00 | 1.00 | 1.00 | 1.00 | 0.94 | 1.00 | 1.00 | 1.00 | 1.00 | 0.98 | 1.00 | 1.00 | 0.90 | 0.78 |
| hammer | 0.84 | 0.80 | 0.64 | 0.98 | 0.96 | 0.58 | 0.82 | 0.68 | 0.92 | 0.94 | 0.40 | 0.48 | 0.48 | 0.62 | 0.52 |
| pick-place | 0.88 | 0.70 | 0.88 | 0.88 | 0.88 | 0.54 | 0.48 | 0.54 | 0.66 | 0.44 | 0.02 | 0.10 | 0.02 | 0.22 | 0.08 |
| push | 1.00 | 0.94 | 0.96 | 0.98 | 0.98 | 0.98 | 0.90 | 0.88 | 0.90 | 0.90 | 0.32 | 0.50 | 0.34 | 0.62 | 0.44 |
| reach | 0.96 | 0.48 | 0.96 | 0.34 | 0.90 | 0.98 | 0.38 | 1.00 | 0.34 | 0.54 | 0.34 | 0.22 | 0.24 | 0.36 | 0.30 |
| window-open | 0.64 | 0.36 | 0.88 | 0.34 | 0.44 | 0.60 | 0.28 | 0.84 | 0.40 | 0.40 | 0.36 | 0.38 | 0.56 | 0.38 | 0.50 |
| Average | 0.82 | 0.72 | 0.79 | 0.77 | 0.83 | 0.72 | 0.70 | 0.73 | 0.72 | 0.71 | 0.45 | 0.50 | 0.51 | 0.52 | 0.46 |

