# OpenReview forum: "Building Generalist Robot Policy from Pre-trained Visual Representations"
_ICLR.cc/2025/Conference — ICLR 2025 Conference Withdrawn Submission_

### Official Review · Reviewer_GfGh · 2024-10-27

**Soundness:** 2
**Presentation:** 3
**Contribution:** 2
**Rating:** 3
**Confidence:** 5

**Summary:**

This paper first investigates the effectiveness of using off-the-shelf, frozen pre-trained vision encoders in a language-conditioned, multi-task robot policy learning setting. The evaluation also considers generalization ability in new visual settings. The encoders include "foundation models" like CLIP and DINOv2 trained on non-robotics images, and models specifically designed for robot tasks like R3M and VC-1. The two findings are 1) spatial feature outperforms the global feature and 2) among all models evaluated, DINOv2 is the best-performing model. However, all models experience a performance drop when visual domain shifts. To merge the gap, this paper investigates different augmentation methods and few-shot adaptation. Results show that DINOv2 still has the best performance. Specific to the few-shot adaptation case, this paper proposes a self-distillation objective to fine-tune the policy network. The self-distillation method improves the performance when the few-shot data is limited and the pre-trained visual encoder is not strong enough.

**Strengths:**

- The study of using pre-trained visual representations for robotics is important as there are more and more visual foundation models. Understanding their performance in policy learning benefits the robotics community.

- This paper particularly investigates the effectiveness of pre-trained visual representations in a generalist setting, which is a meaningful direction.

- Evaluate the representations for in-distribution conditions and unseen conditions such as color and unseen environments.

- Propose the self-distillation objective on latent features in the policy network to perform the few-shot adaptation. The self-distillation is common in other fields of applications, but not that common in policy learning.

**Weaknesses:**

- One of the main findings in this submission is not new, as Shang et al. [1] have shown the importance of using spatial features for Transformer-based visual encoders and the performance scaling regarding model sizes. This submission skips other commonly used pre-trained vision foundation models (in robotics) such as ViT [2], MVP [3](or MAE [4]), SigLIP[5], and recent work like RADIO[6] and Theia [1].

- The paper studies the visual representations in building general policies, but the evaluations are conducted on a very limited scale.
  * First, the evaluations are done in only one simulation suite -- Metaworld.  The number of tasks is not explicitly mentioned in the main paper or the Appendix. From Table 12 in the Appendix, it looks like there are only 10 tasks. This scope is a bit far away from being called a generalist policy.

  * The evaluations should consider other benchmarks with different visual domains. Available simulated benchmarks such as LIBERO [7], CALVIN [8], RoboCasa [9], DMC [10], and Habitat might be helpful (like what Cortexbench organized). I strongly recommend the author also use real-world data like OXE [11] or DROID [12], just like what OCTO [13] or OpenVLA [14] did, where the data show more complex visual distributions.

  * Settings are not clear, such as the number of random runs and amount of demonstration used.


- More importantly, this submission lacks analysis on **why** different visual representations exhibit different performance under **generalist policy learning**. This question is very interesting to me, but unfortunately, I can not find any explanations in the submission. I wish the authors could discuss any in-depth connections between the policy performance and how visual representation/encoder is obtained, such as datasets, objectives, properties, and architectures. It could also be more quantitative measurements you find in your study.

- What's the purpose of inputting multiple views? How do the findings in this submission transfer to other settings with more or fewer views? Within the scope of this submission, I would assume DINOv2 gets the best performance because of its better cross-view alignment performance using its feature. Is that true? The study is missing.

- The value of different augmentation techniques is unclear. Though different techniques have different effects on different models, DINOv2 seems to be the best most of the time. It would be good to further investigate why and how to further improve the best-performing model.

- The proposed few-shot adaptation looks interesting, but the connection to **pre-trained visual representations** is unclear. The self-distillation applies to the latent feature in the **policy network** (towards the head part from my understanding). How does this specific technique address the discrepancy between seen and new visual representations? Why do different pre-trained visual representations exhibit different performances? I also recommend testing LoRA [15] in the few-shot adaptation evaluation. More interestingly, one recent preprint ReVLA [16] claims that resetting the visual encoder weights to the original ones after fine-tuning could benefit generalization. The authors may also consider this direction. I also recommend authors compare to other adaptation methods or continual learning methods to thoroughly investigate the adaptation method.


Overall, I believe the technical findings and evaluations in the current submission are limited, the novelty is somewhat but needs further careful evaluation. The clearness of ideas is good, but not for technical details like environmental settings. Based on these reasons, I recommend a rejection at this moment.

References:

[1] Shang et al., "Theia: Distilling Diverse Vision Foundation Models for Robot Learning", CoRL 2024

[2] Dosovitskiy et al., "An Image is Worth 16x16 Words: Transformers for Image Recognition at Scale", ICLR 2021

[3] Xiao et al., "Masked Visual Pre-training for Motor Control", 2022

[4] He et al., "Masked Autoencoders Are Scalable Vision Learners", CVPR 2022

[5] Zhai et al., "Sigmoid Loss for Language Image Pre-Training", ICCV 2023

[6] Ranzinge et al., "AM-RADIO: Agglomerative Vision Foundation Model - Reduce All Domains Into One", CVPR 2024

[7] Liu et al., "LIBERO: Benchmarking Knowledge Transfer for Lifelong Robot Learning", NeurIPS 2023

[8] Mees et al., "CALVIN: A Benchmark for Language-Conditioned Policy Learning for Long-Horizon Robot Manipulation Tasks", IEEE RA-L, 2022

[9] Nasiriany et al., "RoboCasa: Large-Scale Simulation of Everyday Tasks for Generalist Robots", RSS 2024

[10] Tassa et al., "DeepMind Control Suite", 2018

[11] Open X-Embodiment: Robotic Learning Datasets and RT-X Models, ICRA 2024

[12] DROID: A Large-Scale In-the-Wild Robot Manipulation Dataset, 2024

[13] Ghosh et al., "Octo: An Open-Source Generalist Robot Policy", RSS 2024

[14] Kim et al., "OpenVLA: An Open-Source Vision-Language-Action Model", CoRL 2024

[15] Hu et al., "LoRA: Low-Rank Adaptation of Large Language Models", 2021

[16] Dey et al., "ReVLA: Reverting Visual Domain Limitation of Robotic Foundation Models", 2024

**Questions:**

Please see the weaknesses above.

---

### Official Review · Reviewer_1dUx · 2024-10-28

**Soundness:** 2
**Presentation:** 3
**Contribution:** 2
**Rating:** 3
**Confidence:** 4

**Summary:**

The paper explores how to leverage pre-trained visual representations for simulated robotic manipulation tasks. The authors first investigate the effectiveness of using feature maps over the global features and then propose to further improve the visual representations via data augmentations. Finally, the authors also propose a few-shot adaptation method for efficient imitation learning.

**Strengths:**

1. The experiments are extensively conducted in simulated tasks.
2. The discovery is possibly useful for simulated robot learning.

**Weaknesses:**

1. Lack of real robot experiments. Due to the large visual gap between simulation and the real world, and considering that the focus of this paper is to study the pre-trained visual representations which is pre-trained on real-world data, only simulated experiments can not support the arguments of authors. Besides, the diversity of simulated tasks is also very limited. It would be good if authors show more real robot results that are consistent with simulation results, and more challenging simulation tasks beyond MetaWorld might be good, such as RoboMimic/ManiSkill/RLBench.
2. Lack of novelty. The most interesting takeaway from this paper is the usefulness of local features over global features, which however is mostly obvious and well-known to the community. Besides, even the technical contributions of this paper seem to be very incremental.
3. Overclaim of the title. Though the title is about "building generalist robot policy", I would suggest that authors carefully choose a humble title to better reflect their actual contributions of the paper. An example of the generalist robot policy is [1].

[1] https://www.physicalintelligence.company/blog/pi0

**Questions:**

See weakness. It would be good if the authors provide more diverse experiments in the real world and also carefully select the title.

---

### Official Review · Reviewer_U8VV · 2024-11-03

**Soundness:** 2
**Presentation:** 2
**Contribution:** 2
**Rating:** 3
**Confidence:** 4

**Summary:**

This paper investigates the use of vision pre-trained models (PTMs) for developing generalist robot manipulation policies.

The authors find simply keeping local features from the last layers of PTMs can significantly improve the policy performance compared to the global feature.

The authors also study the effects of conventional data augmentation methods on robot policy training with pre-trained visual representations.

Finally, the authors propose a novel objective for few-shot adaptation by introducing self-distillation on features from a trained policy.

**Strengths:**

The writing and organization of this paper are good.

The simulation experiments on metaworld are solid.

**Weaknesses:**

Many works on robotics vision representation learning have not been mentioned, such as [1-6].
[1] Real-world robot learning with masked visual pre-training.
[2] Masked visual pre-training for motor control.
[3] Language-driven representation learning for robotics.
[4] An unbiased look at datasets for visuo-motor pre-training.
[5] Spatiotemporal Predictive Pre-training for Robotic Motor Control.
[6] Learning Manipulation by Predicting Interaction.

The authors make many findings; however, all experiments are conducted solely in the metaworld simulation environment, lacking real-world experiments.

Although local features are more effective than global features, such a comparison is not fair, as the former tends to generate more computational load. I believe that the use of global features in previous robotics vision representation learning works was to create a fair and simple evaluation baseline. In reality, many end-to-end learned generalist policies utilized local visual features, such as RT2-X, OpenVLA and Octo. Therefore, I don't consider this to be a new discovery.

In section 5.1, why when there are 5 demonstrations available for each task, the fine-tuning dataset  Dft contains 500 samples (5×10 = 50).

From Figure 8, it can be seen that the proposed self-distillation adaptation method does not yield significant advantages. Furthermore, comparing it with other adaptation methods, in addition to end-to-end fine-tuning, would be better and more convincing, such as designing adapters.

**Questions:**

Please see the weaknesses section.

---

### Official Review · Reviewer_1HHD · 2024-11-03

**Soundness:** 1
**Presentation:** 2
**Contribution:** 1
**Rating:** 3
**Confidence:** 5

**Summary:**

This paper investigates the effectiveness of visual backbones for manipulation tasks. They found that global feature are insufficient to train robust robot model, therefore proposed augmentation methods to resolve the issue.

**Strengths:**

This paper conducts extensive study on meta-world benchmark.

**Weaknesses:**

1. Experiments Limited to Simulation without Real-World Validation
The experiments in this study are conducted exclusively in simulation, with no validation on a physical robot. While the work explores visual representations for robotic models, the lack of real-world testing severely limits the relevance of its findings. Given the substantial sim-to-real gap, conclusions drawn solely from simulated environments are unreliable, as these environments are often overly simplified and do not accurately represent real-world conditions.

2. Limitations of Metaworld as a Benchmark
Metaworld is a relatively simple simulation benchmark, even within the realm of simulation-based studies. A significant limitation is its low image resolution, which lacks sufficient detail for robust evaluation. Although the paper does not report image resolution, it is commonly known that Metaworld images are only 112 x 112 pixels. This resolution is inadequate for making meaningful assessments of different visual encoders’ effectiveness.

3. Unreliable Experimental Results
Several observations in this paper contradict prior research. For instance, the authors claim that R3M outperforms other pre-trained models (lines 288–293). However, multiple studies, including [2], ACT, and Diffusion Policy, have found that backbones pre-trained with CLIP or ImageNet are more effective for manipulation tasks.

4. Omission of Numerous Related Works
The paper overlooks a substantial body of relevant literature, such as [1,2,3,4], which focuses on pre-trained visual representations. This oversight suggests a lack of familiarity with key works in this domain.

5. Incomplete Implementation Details
Important details about the implementation are missing. The paper does not specify the number of demonstrations used for training, the number of tasks evaluated, or the performance of methods across varying task difficulties (easy/medium/hard/very hard). Appendix A provides training hyperparameters, but no information on the experimental settings is offered, making it difficult to assess the robustness of the findings.

[1] WHAT MAKES PRE-TRAINED VISUAL REPRESENTATIONS SUCCESSFUL FOR ROBUST MANIPULATION? CoRL 24
[2] On Pre-Training for Visuo-Motor Control: Revisiting a Learning-from-Scratch Baseline, ICML 24
[3] Masked visual pre-training for motor control, CoRL
[4] Robot Learning with Sensorimotor Pre-training, CoRL

**Questions:**

See weakness. The authors should provide detailed implementation specifics and clearly outline the experimental settings used. Additionally, they should discuss differences with related work thoroughly. Where conclusions diverge from prior research, the authors should offer explanations for these discrepancies. To address the sim-to-real gap, experiments on physical robots are necessary. Currently I do not believe this paper contributes meaningfully to the field.

---

### Official Review · Reviewer_bPZT · 2024-11-04

**Soundness:** 2
**Presentation:** 3
**Contribution:** 2
**Rating:** 5
**Confidence:** 4

**Summary:**

This paper explores the potential of using vision pre-trained models (PTMs) to build generalist robot manipulation policies, capable of performing multiple tasks and generalizing to unseen scenarios. The authors mainly investigate the difference between utilizing local and global feature with different pre-trained vision backbones and find that local feature is better. For generalization, the authors investigate different augmentation in both spatial and temperal pattern. Further, the paper designs a self-distillation training framework for few-shot adaptation.

**Strengths:**

Focus on visual representation learning: By focusing on the role of visual PTMs, the paper provides a deeper understanding of what kind of visual information could benefit robot control. This is crucial for developing more efficient and robust robot policies.

Use of diverse evaluation metrics: The research employs a comprehensive set of evaluation metrics, including success rates for in-domain and out-of-domain tasks (unseen colors and unseen environment). This allows for a more nuanced analysis of policy performance and generalizability.

Comparison with state-of-the-art models: The study includes a comparison with state-of-the-art PTMs and demonstrates the superiority of DINOv2 for robot manipulation tasks. This provides valuable guidance for practitioners in choosing the most suitable model for their specific applications.

Open-source code and datasets: The paper mentions the availability of open-source code and datasets, enabling other researchers to replicate the experiments and build upon the findings. This promotes transparency and collaboration within the research community.

Few-shot adaptation: Few-shot adaption is important for the community and the idea of self-distillation is interesting.

Insights into the role of inductive biases: The paper discusses the impact of PTMs’ training objectives and inductive biases on policy learning. This provides valuable insights into how different training approaches can influence the generalization capabilities of robot policies.

**Weaknesses:**

The evaluation benckmark: i think only a single simulator with three views is not enough for getting these conclusions, the authors should introduce another simulator and conduct some real world experiments.

The lack of vision language pretraining: the vision encoder utilized by vision language models (like qwen) should be involved into evaluation, and the authors should testify the performance on unseen language tasks.

The performance gain of self-distillation: comparred with finetuning, the performance gain of self-distillation not enough to demonstrate its efficiency.

**Questions:**

See weakness.

---

### Note · Authors · 2024-11-26

**Comment:**

We sincerely thank all reviewers for thoughtful and constructive feedback on our manuscript. Your insights have been invaluable in improving the clarity, rigor, and overall quality of this work. We deeply appreciate the time and effort you devoted to reviewing our paper. Thank you again for your valuable feedback and for helping us strengthen our manuscript.

**Withdrawal Confirmation:**

I have read and agree with the venue's withdrawal policy on behalf of myself and my co-authors.